# Lipase Production by *Yarrowia lipolytica* in Solid-State Fermentation Using Amazon Fruit By-Products and Soybean Meal as Substrate

Aparecida Selsiane Sousa Carvalho [1], Júlio Cesar Soares Sales [1], Felipe Valle do Nascimento [2],
Bernardo Dias Ribeiro [3], Carlos Eduardo Conceição de Souza [1], Ailton Cesar Lemes [3]
and Maria Alice Zarur Coelho [3,*]

1   Departamento de Bioquímica, Instituto de Química, Universidade Federal do Rio de Janeiro,
    Av. Athos da Silveira Ramos 149, Ilha do Fundão, Rio de Janeiro 21941-909, Brazil
2   Departamento de Tecnologia de Processos Bioquímicos, Instituto de Química,
    Universidade do Estado do Rio de Janeiro, Rua São Francisco Xavier, 524 Pavilhão Haroldo Lisboa da Cunha,
    Rio de Janeiro 20550-013, Brazil
3   Departamento de Engenharia Bioquímica, Escola de Química, Universidade Federal do Rio de Janeiro,
    Av. Athos da Silveira Ramos 149, Ilha do Fundão, Rio de Janeiro 21941-909, Brazil
*   Correspondence: alice@eq.ufrj.br

**Abstract:** The production of polyunsaturated fatty acids from fish oil, which is related to various health benefits including effects against cardiovascular diseases, antihypertensive, anticancer, antioxidant, antidepression, anti-aging, and anti-arthritis effects, among others, can be advantageously performed through the application of lipase. However, the high cost associated with enzyme production can make the process unfeasible and thus alternative substrates should be investigated to solve these problems. This research aimed to produce lipase by *Yarrowia lipolytica* IMUFRJ50682 in solid-state fermentation using by-products of the food processing industry (andiroba oil cake and soybean meal) and verify the potential application in the initial hydrolysis of fish oil to further produce polyunsaturated fatty acids in a suitable process. A screening was carried out for the analysis of andiroba oil cake and soybean meal combinations in different proportions (0:100 to 100:0, respectively) at 48 h of the fermentation process. Afterward, the solid matrix composed by soybean meal and andiroba oil cake was supplemented with soy oil and Tween 80 to improve the lipase activity. The enzymatic extract was characterized in relation to the protein profile by electrophoresis. Finally, the enzymatic extract and the solid biocatalyst produced were applied to evaluate the potential hydrolysis of the fish oil in an initial study. Maximum lipolytic activity (63.7 U·g$^{-1}$) was achieved using andiroba oil cake and soybean meal (50:50) after 24 h of fermentation. Soybean oil 1.5% and Tween 80 (0.001%) in an emulsion provided an increase of 1.5-fold (82.52 U·g$^{-1}$) in the enzyme activity. The electrophoretic analysis demonstrated a band between 37 and 40 kDa that may be related to lipase and a band of 75 kDa referring to the α subunit of the β-conglycinin present in soybean meal. After 48 h, the solid biocatalyst showed a higher degree of hydrolysis (DH) (71.0%) than the enzymatic extract (61.5%). The solid biocatalyst was stable during storage at room temperature for 7 months. The production of lipases using Amazon fruit by-product and soybean meal in solid-state fermentation is viable as well as the application of the extract and solid biocatalyst in the initial application for the hydrolysis of fish oil to further produce polyunsaturated fatty acids in an industrially suited process.

**Keywords:** *Carapa guianensis* Aublet; lipase production; solid-state fermentation; sustainability; *Yarrowia lipolytica*



## 1. Introduction

Amazonian fruits have a high biotechnological potential for research and application in various sectors of the economy such as the pharmaceutical and food industries due

to the presence of macro and micronutrients including proteins, carbohydrates, fibers, carotenoids, polyunsaturated fatty acids, and vitamins, among others [1].

Andiroba (*Carapa guianensis* Aublet) is an Amazonian plant that produces oilseed fruits with commercial potential for oil extraction with medicinal properties as an analgesic, anti-edematogenic, and anti-inflammatory effect [2,3]. In addition, it presents interesting technological properties for the cosmetic industry because it is rich in emollient compounds, being supplied as an input for cosmetic industries around the world [4]. The andiroba fruit is a capsule that weighs between 90 and 540 g and is composed of 1 to 16 brown seeds, weighing between 1 and 70 g. To obtain the oil, the seeds are broken into small pieces and dried in ovens until they reach 8% moisture. Then, they are compressed in metallic expeller-type presses [5], producing the andiroba oil that presents great industrial interest and high commercial value since this product can exceed 50 dollars per liter, depending on its characteristics [6].

The annual production of andiroba is about 122.16 tons [7] and the oil production did not exceed 30% of the weight of the fruit [8]. These numbers mean that approximately 36,648 tons of oil and 85,512 tons of cake are generated annually. The by-product obtained after the exploration of these fruits can still contain lipids (20%), ash (4.2%), proteins (7.3%), and carbohydrates (17%), and because of this composition, its use in biotechnological processes is interesting [9]. One way to apply these by-products is through solid-state fermentation (SSF) to produce high-value-added compounds such as enzymes.

Solid-state fermentation is carried out using a solid matrix in the absence of free water, but the solid must have enough moisture for microbial growth [10]. The use of SSF has been reported to produce aroma compounds [11], antibiotics [12], organic acids [13], biopesticides [14], biosurfactants [15], enzymes [16], and other biocompounds. Enzyme production in SSF requires the use of microorganisms such as fungi and bacteria that grow under conditions commonly used in SSF. Among the microorganisms used biotechnologically, yeast stands out, especially *Yarrowia lipolytica*, due to its potential to produce lipases [10].

*Yarrowia lipolytica* is a non-conventional yeast that has a maximum growth temperature at 32–34 °C and is unable to survive under anaerobic conditions. *Y. lipolytica* is classified as generally recognized as safe (GRAS) by the Food and Drug Administration (FDA) and this enables its safe use in food and pharmaceuticals processes/products [9]. This yeast is recognized as producing intracellular, extracellular, and cell wall-associated lipases. *Yarrowia lipolytica* can produce lipases using a variety of carbon sources including oils, methyl esters, and fatty acids [17]. The mentioned features make *Y. lipolytica* versatile for biotechnology applications, especially in the production of lipases.

Lipases (triacylglycerol acylhydrolases EC 3.1.1.3) are enzymes that catalyze the hydrolysis of triglycerides to free fatty acids and glycerol in the presence of water [18]. Lipases can be applied in many sectors including the food industry, among others. The use of lipases has been reported in the hydrolysis of milk, fat, and cheese ripening, in the modification of butter fats, in the improvement in the aroma of beverages and bakery, in food dressing, and in the flavor development of meat and fish [19]. In addition, lipases have already been applied to oil hydrolysis to the concentration of polyunsaturated fatty acids that can be used in the food industry as an enriched supplement [20].

Several studies have reported the use of waste and by-products in the production of lipase, among them soybean hulls, watermelon peels, cottonseed meal, soybean meal, and andiroba oil cake [9,21–26]. However, the combination of andiroba oil cake and soybean meal in different proportions for lipase production by *Yarrowia lipolytica* has not been reported yet. This research aimed to produce *Yarrowia lipolytica* lipase by solid-state fermentation (SSF) using by-products of the food processing industry (andiroba oil cake and soybean meal) to apply in the hydrolysis of fish oil to further produce polyunsaturated fatty acids.

## 2. Results and Discussion

The strategy adopted to produce lipase by *Yarrowia lipolytica* in SSF using the combination of andiroba oil cake and soybean meal in different proportions was based according to the flowchart shown in Figure 1.

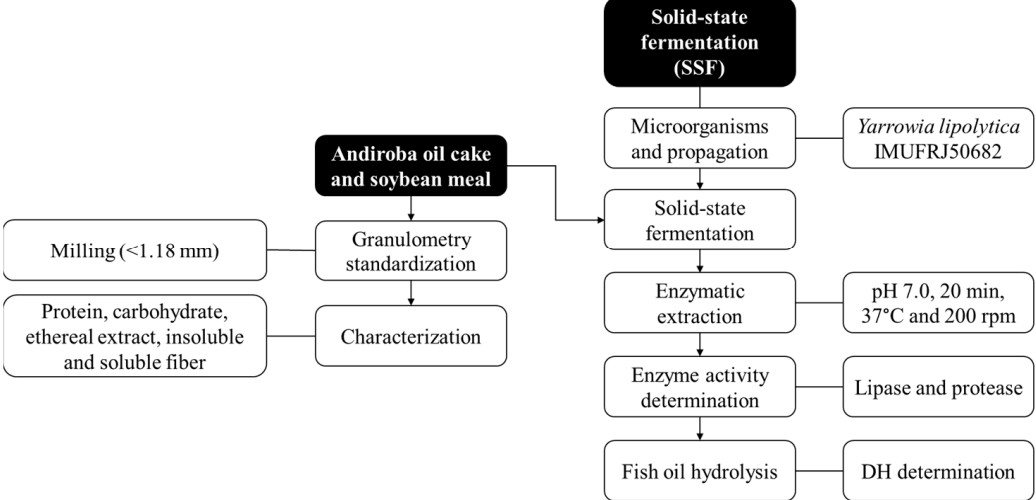

**Figure 1.** Process of lipase production using the combination of andiroba oil cake and soybean meal as solid matrix substrates from *Yarrowia lipolytica* IMUFRJ50682 in SSF.

### 2.1. Agro-Industrial By-Product Characterization

The physical-chemical composition of andiroba oil cake is shown in Table 1. The andiroba oil cake presented 13.7% of proteins, 26.9% of ether extract, 0.4% of carbohydrates, and 45.6% of insoluble fibers. On the other hand, soybean meal presented 48.3% of proteins, 2.4% of ether extract, 14.0% of carbohydrates, and 18.0% of insoluble fibers. Each nutrient present in the cake has a role in microbial metabolism. The nitrogen content is essential for the synthesis of proteins, nucleotides, and secondary metabolites for the growth and metabolism of microorganisms as well as for the biosynthesis of enzymes [25]. The residual oil content (ether extract) present in the andiroba oil cake acts as a carbon source, participating in the formation of biomass and the production of energy for the performance of cellular function [26]. Additionally, the oil present in the andiroba cake is a hydrophobic substrate and can acts as an inducer in the production of lipase, as reported by Brígida et al. [17]. Initially, the lipase bound to the microorganism cell hydrolyzes the lipid present in the medium and initiates cell growth. Along the process, substrate availability decreases, so the organism secretes extracellular lipases [27].

**Table 1.** Physical-chemical composition of andiroba oil cake and soybean meal.

| Component | Andiroba Oil Cake (%) | Soybean Meal (%) |
|---|---|---|
| Proteins | 13.7 ± 1.2 | 48.3 ± 0.64 |
| Ethereal extract | 26.9 ± 0.66 | 2.4 ± 0.04 |
| Carbohydrates | 0.4 ± 0.92 | 14.0 ± 3.68 |
| Ashes | 4.5 ± 0.00 | 6.3 ± 0.05 |
| Insoluble fibers | 45.6 ± 0.95 | 18.0 ± 1.58 |
| Cellulose | 14.4 ± 0.85 | 6.9 ± 0.40 |
| Lignin | 22.7 ± 1.02 | 0.2 ± 0.10 |
| Hemicellulose | 8.5 ± 0.85 | 10.9 ± 0.58 |
| Porosity ($m^3_{air} \cdot m^{-3}_{bed}$) | 0.425 | 0.412 |

The ash quantified in andiroba oil cake (4.5%) and soybean meal (6.3%) represents the content of several minerals and can be used in the metabolism of the microorganism; however, some minerals such as sodium and manganese ions can inhibit *Yarrowia lipolytica* growth [17], and therefore it is necessary to properly evaluate the by-products.

The carbohydrates and fiber contents in raw materials can be used in microbial nutrition in a limited way since *Yarrowia lipolytica* does not produce amylases or cellulases and hemicellulases [28,29], making it difficult to break down these polysaccharides and release carbon-rich compounds to be used in yeast metabolism. Thus, it is suggested that the yeast growth is mainly due to the presence of the hydrophobic substrates in the cake.

The porosity values of the andiroba oil cake and the soybean meal are shown in Table 1. According to Table 1, the porosity of the andiroba cake and the soybean meal were 0.425 and 0.412 $m^3_{air} \ m^{-3}_{bed}$, respectively. Higher porosity values are associated with improved mass and heat transfer during fermentation, leading to improved enzyme production. In addition, porosity is an important parameter to understand the moisture of the medium: high moisture values may be associated with low porosity values, and this affects the aeration of the medium, and consequently the production of lipase [22].

## 2.2. Lipase Production by SSF

The use of by-products for lipase production by solid-state fermentation and the production of stable solid biocatalyst allows for the application of the enzyme directly in the reaction medium, reducing costs associated with the enzyme extraction and purification for subsequent application. Furthermore, the use of by-products provides nutrients for microbial growth and development, acts as physical support for the microorganism, and reduces culture medium costs to the process. In addition, solid-state fermentation allows for the use of residues and agro-industrial by-products, being an alternative to reducing the environmental problems associated with inappropriate waste disposal [30].

After characterization, the by-products were used as a solid matrix substrate in the production of lipase using SSF, being monitored in relation to lipolytic and proteolytic activities, moisture, and pH during the fermentation process. Figure 2 shows the measured response variable profiles over 48 h of fermentation using *Y. lipolytica* and the combination of andiroba oil cake and soybean meal in different proportions as substrates.

Regarding the use of the substrates individually, the maximum lipolytic activity of 4.36 $U \cdot g^{-1}$ was found for soybean meal (0:100; Figure 2A) and 13.48 $U \cdot g^{-1}$ for andiroba oil cake (100:0; Figure 2B), after 12 and 24 h, respectively. After reaching the point of greatest lipase activity, a reduction in its values was observed, which may be related to the increase in protease activity. This enzyme has an affinity for protein structures, and due to this, can act to degrade the lipase structure, reducing its activity [24,31,32].

Andiroba cake and soybean meal used in different proportions of (25:75), (50:50), and (75:25) produced the maximum values of lipase activity of 57.21 $U \cdot g^{-1}$ after 12 h (Figure 2C), 63.70 $U \cdot g^{-1}$ (Figure 2D) and 40.13 $U \cdot g^{-1}$ (Figure 2E) after 24 h of fermentation, respectively, indicating that the mixture of the matrices can improve the lipase activity. The improvement in enzymatic activity after a combination of andiroba oil cake and soybean meal can be explained by observing the physicochemical compositions of both raw materials, as shown in Table 1. Soybean meal has a high protein content while andiroba oil cake has a higher oil content. Thus, the mixture of andiroba oil cake and soybean meal for lipase production proved to be more interesting than using each one separately, and this favors the non-dependence of a single substrate as well as the use of more than one by-product.

The initial pH of the matrices is acidic due to the presence of free fatty acids present in the oleaginous matrix, and this condition favors *Yarrowia lipolytica* growth since this yeast needs a lightly acid medium for good growth [33], avoiding a medium buffering step. There is also a gradual increase in pH along the process in the matrices ranging from 6.39 to 7.32 (Figure 2A), 5.69 to 6.82 (Figure 2C), 5.64 to 7.68 (Figure 2D), and 5.21 to 6.03 (Figure 2E). In the matrix containing only andiroba oil cake, a gradual decrease in pH (4.8 to 4.26, Figure 2B) was observed due to the oil residual content in the solid medium since the oil is initially hydrolyzed by lipase, not being consumed by the yeast in the initial hours, thus maintaining a slightly acidic pH.

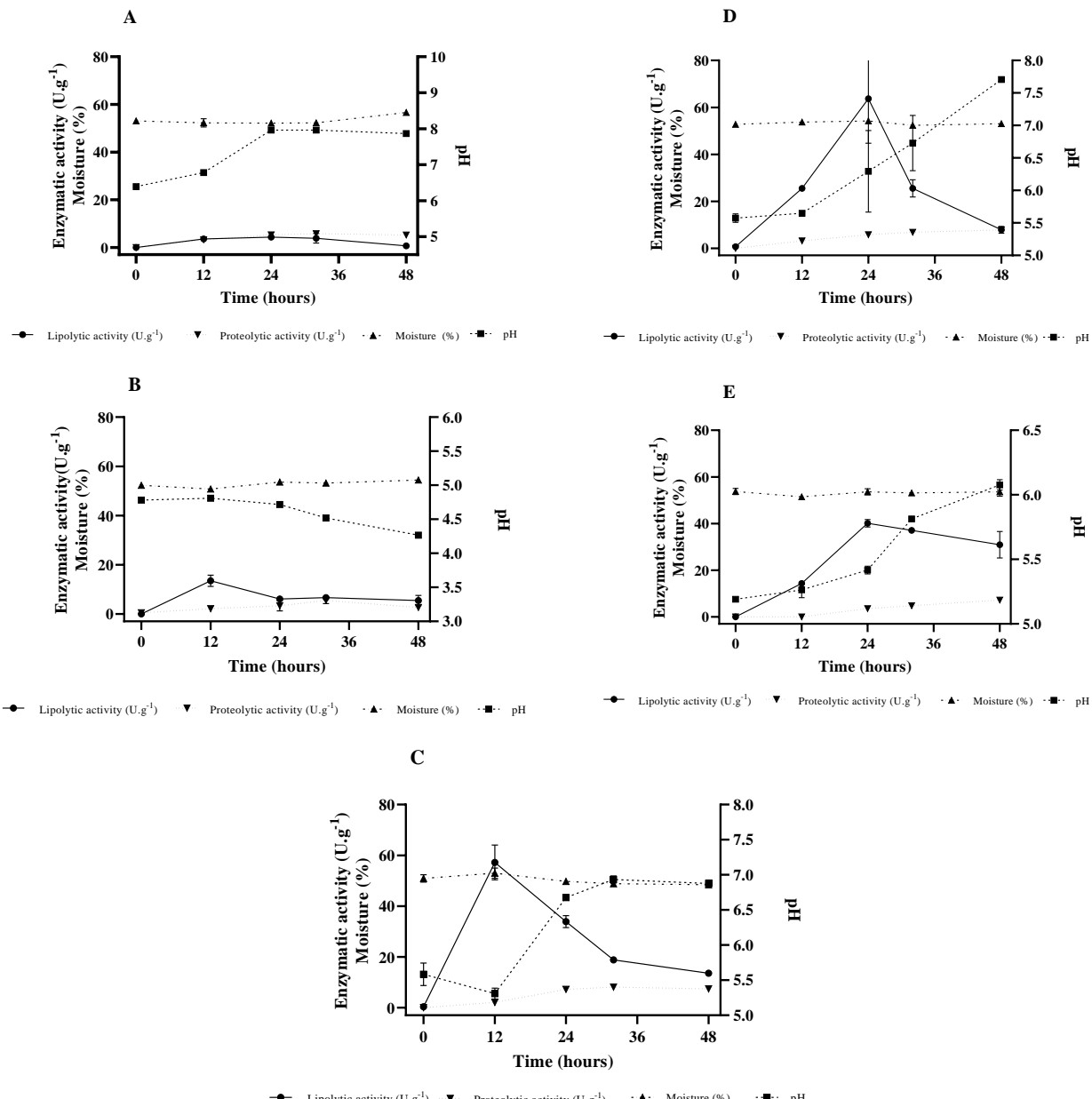

**Figure 2.** Screening of the matrices for the production of lipase in solid-state fermentation by *Yarrowia lipolytica*. (**A**) (0:100), (**B**) (100:0), (**C**) (25:75), (**D**) (50:50), and (**E**) (75:25). The proportion presented refers to the concentration of andiroba oil cake and soybean meal, respectively (andiroba:soybean).

Regarding the moisture content, no variations were observed throughout the fermentation, demonstrating that the incubation system with saturated moisture and the matrices were effective at retaining the moisture. A fermentation medium able to maintain an optimal moisture content is essential for lipase production by *Yarrowia lipolytica* [24], since this parameter can affect the aeration of the system. Furthermore, the ideal moisture content is required to provide the water necessary for the metabolism of the microorganism and to solubilize the nutrients. When the moisture content is above ideal, a poor diffusion of oxygen may occur in the system, being outside the ideal range for the microorganism [34].

Some studies have reported the use of by-products as substrates in solid-state fermentation. However, in these studies, there is an incorporation of pure carbon and/or nitrogen sources such as glucose, yeast extract, peptone, and urea [35–37] in quantities that can increase the process costs. The ideal matrix must present some characteristics for the application that include acting as a support, providing the necessary nutrients for the mi-

croorganism growth, and the secretion of the biotechnological product of interest. Finding all of these characteristics in a by-product can be arduous, which makes it difficult to find an adequate system for a fermentation process; thus, the search for new raw materials and their combination must be continuous [16].

Thus, the proportion of 50:50 between andiroba cake and soybean meal was chosen for further tests. The choice of this condition is associated with valorizing an unexplored waste from the Amazon region.

### 2.3. Fermentation Matrix Supplementation

In the second part of this work, andiroba oil cake and soybean meal in the proportion of 50:50 was supplemented with 1.5% soybean oil, as shown in Figure 3. The supplementation was carried out to verify if the availability of a more easily assimilated carbon source (soybean oil) improved lipase production in relation to the residual oil content present in the andiroba oil cake. The maximum lipase activity of 56.32 $U \cdot g^{-1}$ was obtained after 28 h of fermentation and the protease activity ranged from 2.71 to 7.60 $U \cdot g^{-1}$. The pH ranged from 5.63 to 7.63 and the moisture remained without major fluctuations throughout the period (53 to 58%). As previously presented, the lipolytic activity obtained after supplementation was lower than the value found without supplementation. Matrix supplementation for lipase production was studied by Farias et al. [38] and Souza et al. [24]. The authors found a lipase activity of 139 $U \cdot g^{-1}$ and 93.9 $U \cdot g^{-1}$ using sludge and soybean oil, respectively. However, when Souza et al. [24] performed fermentation using only soybean meal (without supplementation), the authors found $9.4 \pm 0.3$ $U \cdot g^{-1}$ of lipase activity after 10 h of fermentation, demonstrating that supplementation increased the lipase activity in the studied fermentation matrix.

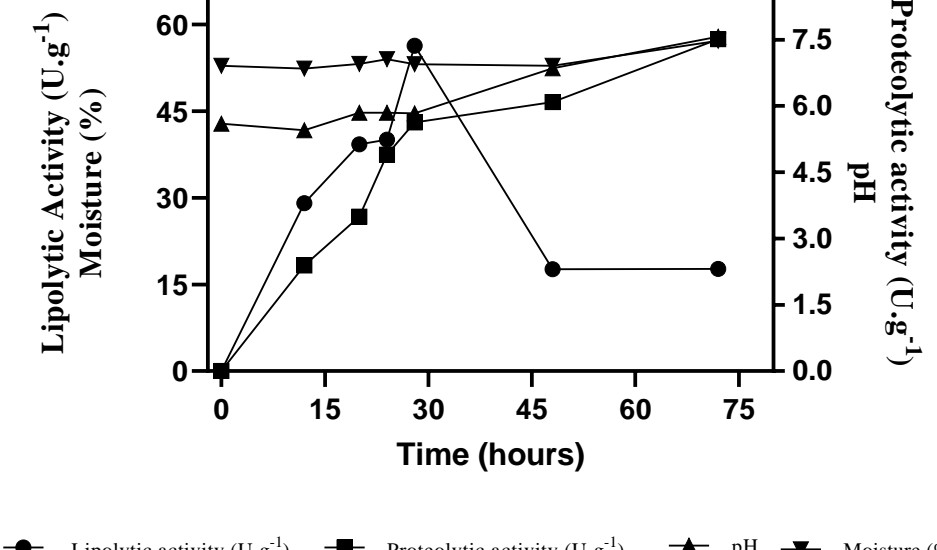

**Figure 3.** Enzyme activity and cultivation conditions of the matrix andiroba oil cake and soybean meal (50:50) supplemented with 1.5 (% *w/v*) soybean oil.

Another component commonly reported for medium supplementation is Tween 80. This is constantly reported in the literature, since this component may increase lipase activity when used in submerged fermentation [39], but there are no reports using Tween 80 in solid-state fermentation. Tween 80 can increase extracellular lipase production during the fermentation process due to the emulsification capacity of the substrate, and therefore improve the substrate accessibility for the microorganism as well as alter cell permeability [40]. Thus, a study was performed combining soybean oil 1.5 (% *w/v*) and Tween 80 0.001% (*w/v*) in an emulsion to supplement the fermentation solid medium, as shown in Figure 4.

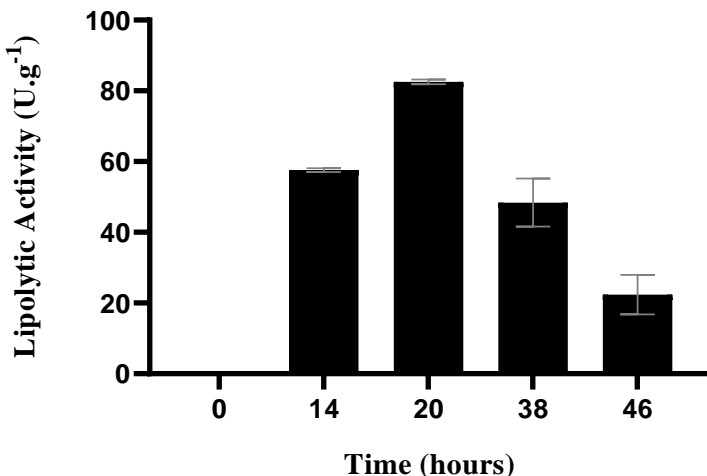

**Figure 4.** Supplementation of the fermentation matrix with soybean oil (1.5% *w/v*) and Tween 80 0.001 (% *w/v*).

In relation to the supplementation with Tween 80, it was verified that after 14 h of fermentation, the lipolytic activity was 57.6 U·g$^{-1}$. The same value was previously obtained only with soybean oil supplementation at 28 h of fermentation, which prolongs the fermentation process and impacts viability. The maximum lipolytic activity after 20 h of fermentation was 82.52 U·g$^{-1}$, a 1.46-fold increase over lipase production using soybean oil alone and an 8-h reduction in the time process.

Long fermentation times increase the cost of the process, so the search for an ideal binomial microorganism—a medium that produces lipase in a shorter time (high productivity) and that uses by-products as matrices is interesting. Several studies in the literature have reported the production of lipase via SSF to achieve adequate conditions and make it feasible to obtain lipases. The use of different raw materials including soybean meal, canola, watermelon peels, and palm kernel oil cake was observed, with activities ranging from 9 to 127 U·g$^{-1}$ [9,22,24].

Sales, Castro, Ribeiro, and Coelho [22] studied lipase production by *Y. lipolytica* in solid-state fermentation using soybean meal supplemented with watermelon peel (5%) obtained an improvement in lipase activity by 31%, a value corresponding to 75.22 U·g$^{-1}$. Souza et al. [41] produced lipase from *Y. lipolytica* by solid-state fermentation for application in the synthesis of commercial esters with value for the food industry, and the results showed that the best fermentation condition to produce the biocatalyst was using soybean meal and soybean oil (3% *w/w*). The enzyme produced under these conditions allowed for a conversion of up to 92.9 to synthesize the useful high-added value esters applied in the food industry.

In addition to the high enzymatic activities obtained in the present work, we can highlight the additional advantages: (i) the possibility of using by-products with low exploitation, with sustainability appeal for the Amazon region; (ii) the possibility of a combination of different by-products, which allows for nutritional complementation for microbial metabolism; (iii) the reduction in pressure on components already widely used in the bioprocess industry such as soybean meal; (iv) economic valorization of by-product producing regions; (v) expansion of the substrate options for lipases production, and (vi) a reduction in the operating costs, among other advantages.

## 2.4. SDS-PAGE

The crude enzymatic extract obtained after fermentation using a matrix composed of andiroba oil cake and soybean meal (50:50) without supplementation was analyzed in SDS-PAGE. The electrophoretic analysis showed the protein bands contained in the enzymatic extract, as presented in Figure 5.

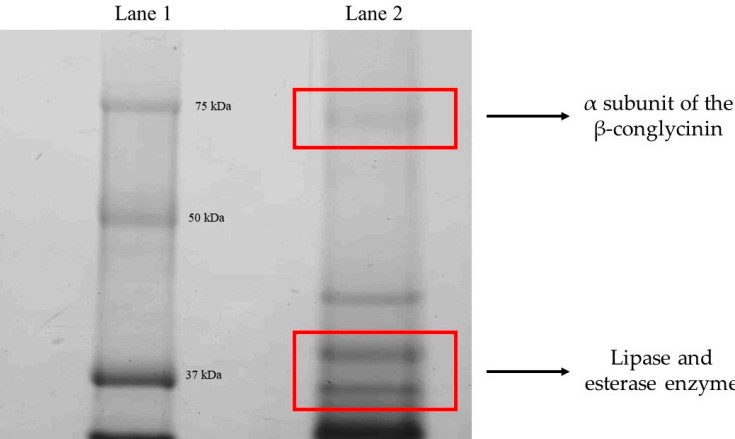

**Figure 5.** Electrophoresis analysis of the crude enzymatic extract (SDS-PAGE) from solid-state fermentation using andiroba oil cake and soybean meal (50:50) without supplementation. Lane 1: Standard protein markers. Lane 2: Andiroba oil cake and soybean meal (50:50) samples.

Lane 1 refers to standard protein markers and Lane 2 is the sample obtained from the combination of andiroba oil cake and soybean meal in the better proportion studied (50:50). It is possible to visualize a band around 37–40 kDa in the crude enzymatic extract that may be related to lipase and/or esterase produced by *Yarrowia lipolytica*, as observed by Souza et al. [41], which may also be the main extracellular lipase of *Y. lipolytica* (Lip2), which was widely produced in all of our studies. Furthermore, it is reported that even for enzymes that are mainly associated with the cell wall (Lip7 and Lip8) of *Y. lipolytica*, the same molecular mass has been observed (37–41 kDa) [42].

Additionally, according to results previously published by our research group, the same bands presented here were related to esterase and lipase activity from the zymogram revealed by α-naphthylyl acetate as the substrate [41], which allowed us to correlate them. The bands of 75 kDa refer to the α subunit of the β-conglycinin present in soybean meal as found by Sales et al. [22] and Cheng et al. [43] in the soybean meal fermentation.

*2.5. Fish Oil Hydrolysis*

Lipases have been used in the hydrolysis of fatty acids to concentrate polyunsaturated fatty acids (PUFAs) [44,45]. The main advantage of the application of lipases in the production of polyunsaturated fatty acids is the specificity of the enzyme and reactions occurring under moderate temperature conditions, which favors the maintenance of the structure of PUFAs [44]. The use of lipases is preferred to chemical methods since they provide glycerides of low yield and purity [46]. The role of lipases in the selective hydrolysis of saturated fatty acids (SFAs) and monounsaturated fatty acids (MUFAs) from triacylglycerols (TAGs) is to produce glycerides rich in PUFAs. The principle of this method is the steric hindrance caused by the molecular configuration of the carbon-*cis* double bonds in PUFAs that cause the folding of fatty acid chains. Thus, the enzymatic active sites do not access the ester bonds of these fatty acids with their glycerol skeletons [47,48]. Numerous benefits are associated with the insertion of fatty acids in the diet such as child development, prevention of cardiovascular diseases, cancer, and various mental disorders (depression, attention deficit disorder, hyperactivity), in addition to the anti-inflammatory potential and potential hypertension control [49].

The production of enzyme extracts to be applied in processes such as hydrolysis can be an expensive task. Thus, researching raw materials that reduce the cost of production can provide an interesting alternative. The use of a solid biocatalyst is desirable because this solid biocatalyst can be reused in enzymatic reactions, in addition to having excellent storage stability at room temperature, without refrigeration costs, and ease of transportation [41]. There have been no reports on the use of a solid biocatalyst from *Yarrowia lipolytica*

to produce PUFAS by the enzymatic hydrolysis of fish oil. Thus, the application of the crude enzymatic extract and solid biocatalyst produced using andiroba oil cake and soybean meal (50:50) was studied to evaluate the potential application of enzymes in the hydrolysis of fish oil to further produce polyunsaturated fatty acids in a suitable process (Figure 6). It is possible to observe a high degree of hydrolysis (DH) in shorter reaction times using a solid biocatalyst (63, 70.8, 72.5, and 74.7%) than the enzymatic extract (47.5, 61.5, 66.5, and 74.8%) after 24, 48, 72 h, and 144 h, respectively.

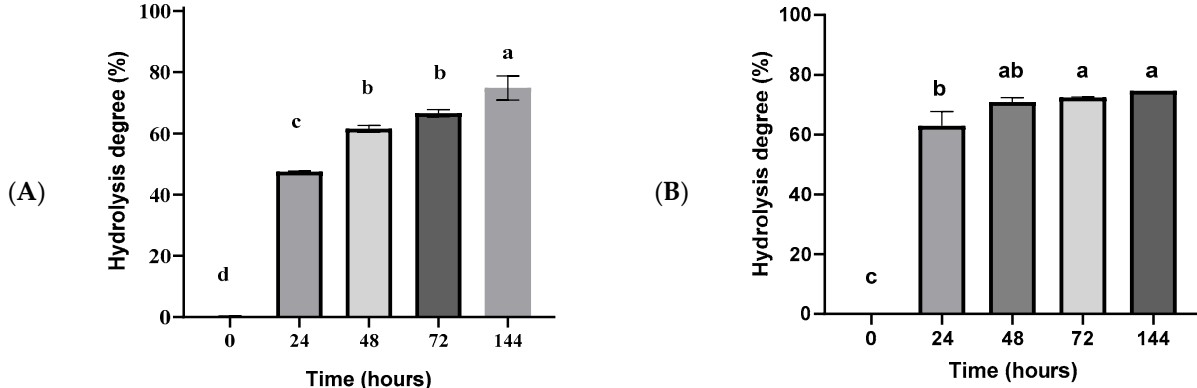

**Figure 6.** Hydrolysis of fish oil after the application of the crude enzymatic extract (**A**) and using the solid biocatalyst stored for 7 months at room temperature (**B**), both produced by *Yarrowia lipolytica*. Data followed by the same lowercase letters are not significantly different ($p > 0.05$).

The enzymatic hydrolysis process is constantly applied to obtain concentrated polyunsaturated fatty acids. Gao et al. [50] used lipase in the hydrolysis of codfish oil and the contents of EPA and DHA were improved 3.24-fold and 1.98-fold, respectively. Aarthy et al. [20] used concentrated lipase (1000 U/mL) in the hydrolysis of fish oils and also found a hydrolysis rate above 60% after 72 h. In this work, better hydrolysis degrees were achieved using a crude enzymatic extract of lipase (i.e., without purification) when comparing the same hydrolysis time. Other authors have studied the hydrolysis of Musteleus mustelus liver oil and seal blubber oil reporting 75% and 70% of hydrolysis after 24 and 9 h of reaction, respectively [25,26].

Martins et al. [51] used a commercial lipase from *Burkholdeira cepacia* (Amano) for the hydrolysis of fish oil and after 48 h of reaction, obtained 55.6% of DHA compared to the maximum calculated content. In our work, in a preliminary study, we obtained 70.8% of hydrolysis after 48 h of reaction using the solid biocatalyst.

Thus far, such findings have suggested the viability of using by-products to produce lipase in solid-state fermentation as a way of mitigating environmental damage, valuating by-products and cost-effectiveness. In addition, the results show the potential application of the lipase enzyme in the hydrolysis of fish oil to further produce polyunsaturated fatty acids in a suitable process.

Despite being abundant in low-cost agro-industrial by-products, the determination and standardization of composition, in addition to cost estimation for obtaining the enzymatic extract and solid biocatalyst from low-cost agro-industrial by-products is still a challenge, but extremely dependent on the type of by-product, seasonality, and quantity generated as well as the process used, and geographic location, among other factors. Thus, issues such as the complexity of the chain and its logistical costs, the use of complex and costly processes, high energy consumption, and regulatory issues, among others, must be overcome. In this sense, the processing of by-products must overcome several barriers before becoming economically viable including the need to process large quantities of raw materials, the capacity to process heterogeneous raw materials, integrated logistics with different processing industries, and the possibility of the integration process in the processing unit to allow for the generation of high-value ingredients, among others [52–55].

## 3. Materials and Methods

### 3.1. Material

Soybean meal was purchased from Caramuru Alimentos (Goiás, Brazil). Andiroba oil cake produced from oil extraction was provided by Beraca Ingredientes Naturais (Pará, Brazil). Both substrates were standardized in relation to granulometry (<1.18 mm) and properly stored under refrigeration in polypropylene packages until use. The fish oil was purchased from Mundo dos Óleos, and according to the manufacturer, it is extracted by cold pressing and filtration, obtained from raw material with guaranteed origin. All other chemicals used were of analytical grade and used as received without any further purification, being obtained from Tedia (acetone), Sigma-Aldrich (St. Louis, MO, USA, glucose, azocasein, agar, yeast extract, ethanol, methanol), Vetec (Tween 80), Oxoid (peptone), Isofar (sodium hydroxide, gum Arabic), and Precision Plus Protein Kaleidoscope—Bio-rad (molecular mass markers, kDa).

### 3.2. Microorganism and Inoculum Cultivation Conditions

*Yarrowia lipolytica* IMUFRJ50682, isolated from an estuary in Guanabara Bay, Rio de Janeiro, Brazil [56] was cultivated at 28 °C in YPD—agar medium (*w/v*: yeast extract 1%; peptone, 2%; glucose, 2%; agar, 3%). The cells were grown in a liquid medium containing yeast extract 1% (*w/v*), peptone, 2% (*w/v*), and glucose 2% (*w/v*) during 72 h, 160 rpm at 28 °C.

### 3.3. Agro-Industrial By-Product Characterization

Physical-chemical composition of the soybean meal and andiroba oil cake was determined in terms of moisture, protein, carbohydrate, ashes, ether extract, insoluble fiber, and soluble fiber content, according to the methodology reported by the Association of Official Analytical Chemists (AOAC) [57]. Additionally, as lipase production by *Y. lipolytica* is affected by aeration [58], the bed porosity in SSF was evaluated according to Equation (1), where $\varepsilon$ is the porosity (expressed in $m^3_{air} \cdot m^{-3}_{bed}$); $\rho drysolid$ is the apparent density of the dry sample ($kg \cdot m^{-3}$); and $\rho wetsolid$ is the density of the sample after water addition ($kg \cdot m^{-3}$) [58].

$$\varepsilon = \left(1 - \frac{\rho\ dry\ solid}{\rho\ wet\ solid}\right) \tag{1}$$

### 3.4. Lipase Production by SSF

The solid matrix containing soybean meal and andiroba oil cake was prepared before the inoculation in the tray-type reactor with different proportions of the substrate and autoclaved at 121 °C for 20 min. The fixed process parameters used for lipase production were moisture of 55% and inoculum concentration of 0.71 mg dry biomass/g substrate [24]. The reactors were incubated in a Biochemical Oxygen Demand (BOD) chamber at 28 °C and sacrificial samples (i.e., one tray-type reactor for sampled time) were taken throughout the fermentation for analysis.

SSF was evaluated using different combinations of andiroba oil cake and soybean meal (0:100; 25:75; 50:50; 75:25 and 100:0) at different times (0, 12, 24, 32, and 48 h). Afterward, the supplementation of the solid matrix containing andiroba oil cake and soybean meal (50:50) was evaluated adding 1.5 (% *w/v*) soy oil over time (0, 12, 14, 20, 24, 28, 48, and 72 h) to obtain an increase in lipolytic activity. In addition, the presence of Tween 80 (0.001% *w/v*) in the fermentation medium containing 1.5 (% *w/v*) soybean oil was tested. Fermentation was monitored by determining the lipase and protease activity as well as the moisture and pH (described in Subsection "3.6. Analytical determinations").

### 3.5. Enzyme Extraction and Production of Solid Biocatalyst

The enzyme extraction was performed by adding 50 mL of 50 mM potassium phosphate buffer pH 7.0 in the bioreactors followed by incubation at 37 °C, 200 rpm, for 20 min. Subsequently, the fermented material suspended in the buffer was pressed using a masher with gaze and centrifuged at 3000 rpm for 5 min. The solid biocatalyst was obtained from

the freeze-drying of the whole mass obtained at the end of the fermenting process for 72 h and stored at room temperature for 7 months to verify the enzymatic stability.

### 3.6. Analytical Determinations

#### 3.6.1. Lipase Activity

Lipase activity was performed using the method proposed by Freire et al. [59]. The reaction medium was emulsified in an Ultra Turrax (IKA) homogenizer using 5% (*w/v*) olive oil and 5% (*w/v*) gum Arabic in 100 mM phosphate buffer (pH 7.0). Enzymatic extract (1 mL) or 0.5 g of the solid biocatalyst was added to 19 mL of the reaction mixture and incubated for 20 min, 200 rpm at 37 °C. The reaction was interrupted by the addition of 20 mL of acetone–ethanol solution and the free fatty acids were titrated in an automatic titrator (Metrohm 916—Ti-Touch) using 0.04 mol/L NaOH solution. One unit of lipase activity (U) was defined as the amount of enzyme that produces 1 μmol of fatty acid per minute, under the assay conditions.

#### 3.6.2. Protease Activity

Protease activity was quantified according to the methodology by Charney and Tomarelli [60]. Enzymatic extract (0.5 mL) was added in 0.5 mL of 0.5% (*w/v*) azocasein solution prepared with 50 mM acetate buffer (pH) and incubated at 32 °C for 40 min. The reaction was stopped by the addition of 0.5 mL of tri-chloroacetic acid solution 15% (*w/v*) and the samples were centrifuged at 3000 rpm for 15 min. The supernatant (100 μL) was added in a 96-microtiter plate containing 100 μL of 5 M potassium hydroxide, and the absorbance at 428 nm was measured in a microtiter plate reader (SpectraMax, Molecular Devices). One activity unit was defined as the amount of enzyme capable of promoting a unitary increase in absorbance per minute.

#### 3.6.3. Moisture Content and pH

Moisture content was monitored using a moisture analyzer balance (AND MX-50). The pH was measured on a pH meter (TECNAL, model TR-107 PT100, Brazil).

### 3.7. SDS-PAGE

The electrophoresis was performed according to the method reported by Laemmli [61] in a polyacrylamide gel (5% stacking, 15% separating, 0.75 mm thickness). The samples were mixed in a ratio (1:4) from a combination of andiroba oil cake and soybean (50:50) with sample buffer containing β-mercaptoethanol, heated at 95 °C for 5 min, and applied on the gel. Electrophoresis was performed at 150 V for 30 min (Bio-Rad, Hercules, CA, USA), and the gel was revealed using Coomassie Blue R-250. A standard protein marker (Bio-rad, Hercules, CA, USA) with molecular weight ranging from 10 to 250 kDa was used.

### 3.8. Fish Oil Hydrolysis: A Potential Application

The degree of hydrolysis (DH) of fish oil was measured by weighing 1 g of fish oil and adding 25 mL of phosphate buffer pH 7.0 to verify the potential application of the enzyme in the hydrolysis of fish oil. Then, 5 mL of the enzymatic extract (37 U) in amber flasks were agitated for 168 h. The reaction was stopped with 20 mL of acetone and the free fatty acids were titrated in an automatic titrator with 0.1 M methanolic KOH. The reaction blank was obtained with the addition of the enzyme only at the end of the reaction.

The degree of hydrolysis (DH) was calculated according to Equation (2):

$$\text{DH} (\%) = \left(\frac{\text{As} - \text{Aa}}{\text{Si} - \text{As}}\right) \times 100 \qquad (2)$$

where As is the sample acidity; Aa is the acidity from autohydrolysis; Si is the saponification index.

*3.9. Statistical Analysis*

All experiments were replicated three times. In each replication, the analyses were conducted in triplicate. Results corresponded to the mean $\pm$ standard deviation. Data were analyzed by the one-way analysis of variance (ANOVA) whereas the Tukey's test ($p < 0.05$) was used to test differences between means using the Sisvar 5.6.

## 4. Conclusions

The fermentation medium obtained after mixing andiroba oil cake and soybean meal was very effective in lipase production. The chosen fermentation matrix was the mixture of andiroba oil cake and soybean meal in a 50:50 ratio, producing 63.70 $U \cdot g^{-1}$ of lipolytic activity. The maximum lipolytic activity was obtained (82.52 $U \cdot g^{-1}$) after using the andiroba oil cake and soybean meal ratio of 50:50 after supplementation with Tween 80 (0.001%) and soybean oil (1.5%). In the electrophoretic analysis, bands of proteins already reported in the literature as YL Lip2 (37 and 40 kDa) were detected. The previous application of lipase in oil hydrolysis provided up to 63% of hydrolysis after 24 h. This study showed that it is possible to produce lipase using by-products from the Amazon region combined with soybean meal and apply it to fish oil hydrolysis to further produce polyunsaturated fatty acids in a suitable process.

**Author Contributions:** Conceptualization, B.D.R., A.C.L. and M.A.Z.C.; Methodology, A.S.S.C., J.C.S.S., F.V.d.N., C.E.C.d.S., B.D.R., A.C.L. and M.A.Z.C.; Formal analysis, A.S.S.C., J.C.S.S. and F.V.d.N.; Investigation, A.S.S.C., J.C.S.S. and F.V.d.N.; Resources, A.S.S.C., J.C.S.S. and F.V.d.N.; Data curation, A.S.S.C., J.C.S.S. and F.V.d.N.; Writing—original draft preparation, A.S.S.C., J.C.S.S., F.V.d.N. and C.E.C.d.S.; Writing—review and editing, B.D.R., C.E.C.d.S., A.C.L. and M.A.Z.C.; Supervision, B.D.R., C.E.C.d.S. A.C.L. and M.A.Z.C.; Project administration, B.D.R., A.C.L. and M.A.Z.C.; Funding acquisition, B.D.R., A.C.L. and M.A.Z.C. All authors have read and agreed to the published version of the manuscript.

**Funding:** This research received no external funding.

**Data Availability Statement:** Not available.

**Acknowledgments:** The authors acknowledge the Coordenação de Aperfeiçoamento de Pessoal de Nível Superior—Brasil (CAPES—Finance Code 001); the Conselho Nacional de Desenvolvimento Científico (CNPq); and the Fundação Carlos Chagas Filho de Amparo à Pesquisa do Estado do Rio de Janeiro (FAPERJ).

**Conflicts of Interest:** The authors declare no conflict of interest.

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
