# Peer review of "Lipase Production by Yarrowia lipolytica in Solid-State Fermentation Using Amazon Fruit By-Products and Soybean Meal as Substrate"

_catalysts, doi:10.3390/catal13020289_

Round 1
Reviewer 1 Report
In this paper, Amazon fruit by products and soybean meal are used as raw materials, and Yarrowia lipotica is used to produce lipase through solid fermentation, and the lipase is used for hydrolysis of fish oil. The research has application prospects, it is suitable for publish on catalysts after revised.
1. Figure 5 is very unclear. It is recommended to add a band includes andiroba oil cake and soybean meal before fermentation and mark the position of lipase on the figure.
2. “2.2 to 2.3” It highly is recommended to compare the work of this paper with the very relevant work published by the author (references 22 and 41) to highlight the characteristics and advantages of this work.
3. The work “2.5. Fish oil hydrolysis” is a little superficial. In the results, only the degree of hydrolysis is checked, but no fatty acid composition is detected. Moreover, the degree of hydrolysis is not only related to the reaction conditions such as temperature, but also closely related to the substrate specificity of the enzyme. It is suggested to add relevant content. It would be better if this part could be compared with other lipase hydrolysis of fish oil.
4. The main work of this paper is to produce lipase by solid-state fermentation. Hydrolysis of fish oil is only a small point. It is suggested that the topic of this paper be properly adjusted.
Author Response
Dear Reviewer,
Thank you for your useful comments and suggestions on our manuscript. We have modified the manuscript accordingly and the detailed corrections are listed below. The manuscript was amended following the recommendations of each reviewer. The changes are marked with different text colors to facilitate visualization.
Reviewer comments:  
Reviewer 1
In this paper, Amazon fruit by-products and soybean meal are used as raw materials, Yarrowia lipolytica is used to produce lipase through solid fermentation, and lipase is used for the hydrolysis of fish oil. The research has application prospects, it is suitable for publishing on catalysts after revision.
Comment: Figure 5 is very unclear. It is recommended to add a band that includes andiroba oil cake and soybean meal before fermentation and mark the position of lipase on the figure.
Response: Thanks for the suggestion regarding the protein profile (SDS-page). We inserted in Figure 2 the indication of the Lanes and proteins detected in the crude extract, including the indication of lipase and esterase, as well as the protein corresponding to the α subunit of the β-conglycinin present in soybean meal (Lines: 275-276).
Regarding the proteins of andiroba meal and soybean meal, we performed additional and preliminary gel electrophoresis and identified several bands, however, without adequate resolution. The nature of the material and its solubilizing properties may have affected proper identification in the non-fermented medium. In relation to soy protein isolated and soybean meal, there are several works published about the profile and properties of the present proteins. However, for andiroba proteins, no work so far presents the protein profile. In future work, the protein profile can be better identified for understanding the material, considering its adequate extraction.
Supplementary figure (Check in uploaded file)- Additional and preliminary gel electrophoresis of lipase enzyme produced from Yarrowia lipolytica in solid-state fermentation using Amazon fruit by-products and soybean meal as substrate. Lane 1: Standard Protein Markers; Lane 2: andiroba oil cake and soybean meal before fermentation
Comment: “2.2 to 2.3” It highly is recommended to compare the work of this paper with the very relevant work published by the author (references 22 and 41) to highlight the characteristics and advantages of this work.
Response: Thanks for the comment. We include a sentence with information from references 22 and 41 to highlight the characteristics and advantages of this work (Lines: 253-261).
Comment: The work “2.5. Fish oil hydrolysis” is a little superficial. In the results, only the degree of hydrolysis is checked, but no fatty acid composition is detected. Moreover, the degree of hydrolysis is not only related to the reaction conditions such as temperature but also closely related to the substrate specificity of the enzyme. It is suggested to add relevant content. It would be better if this part could be compared with other lipase hydrolysis of fish oil.
Response: We appreciate the suggestion and would like to clarify that the results we present in item 2.5 (Fish oil hydrolysis) are the initial results of research that is in progress. This study aimed to produce lipase by Yarrowia lipolytica IMUFRJ50682 in solid-state fermentation using by-products of the food processing industry (andiroba oil cake and soybean meal) and verify the potential application in the initial hydrolysis of fish oil to further produce polyunsaturated fatty acids in a suitable process. In the next steps, the identification of the polyunsaturated fatty acids produced will be performed. We insert as recommended the comparison of fish oil hydrolysis using lipase produced in our study with another commercial lipase (Lines: 338-341).
Comment: The main work of this paper is to produce lipase by solid-state fermentation. Hydrolysis of fish oil is only a small point. It is suggested that the topic of this paper be properly adjusted.
Response: We removed "application in fish oil hydrolysis" from the title (Lines: 2-4) and we specify that the application in the hydrolysis of fish oil is a potential application that should be investigated in future work (Lines: 25-26; 31-32; 40-42; 320-321; 344-346; 449; 451-452).

Reviewer 2 Report
The manuscript entitled "Lipase production by Yarrowia lipolytica in solid-state fermentation using Amazon fruit by-products and soybean meal as substrate: Application in fish oil hydrolysis" describes the production of lipase from the yeast Yarrowia lipolytica using a solid-state process fermentation in the presence of by-products of the food processing industry: andiroba oil cake and soybean meal aiming at the hydrolysis of fish oil to produce polyunsaturated fatty acids. The manuscript is well written, and the experimental work was carried out with full scientific rigor. Some issues must be reviewed before publication on Catalysts, such as:
(i): p. 5, line 179, figure 2: Clarify that the proportion in A to E refers to andiroba cake and soybean meal.
(ii): p. 8, line 275: Change "conformation" to "configuration".
(iii): p. 8, line 275: "cis" should be italicized.
(iv): p. 8, figure 6: Clarify in figure 6 the meanings of the letters "a", "b" "ab" and "c'.
(v): p. 9, item 3.3, eq. (1): Clarify the meanings of "ε" and "ρ".
(vi): p. 9, line 341: What is the meaning of the acronym BOD?
Author Response
Dear Reviewer,
Thank you for your useful comments and suggestions on our manuscript. We have modified the manuscript accordingly and the detailed corrections are listed below. The manuscript was amended following the recommendations of each reviewer. The changes are marked with different text colors to facilitate visualization.
Reviewer comments:  
Reviewer 2
The manuscript entitled "Lipase production by Yarrowia lipolytica in solid-state fermentation using Amazon fruit by-products and soybean meal as substrate: Application in fish oil hydrolysis" describes the production of lipase from the yeast Yarrowia lipolytica using a solid-state process fermentation in the presence of by-products of the food processing industry: andiroba oil cake and soybean meal aiming at the hydrolysis of fish oil to produce polyunsaturated fatty acids. The manuscript is well written, and the experimental work was carried out with full scientific rigor. Some issues must be reviewed before publication on Catalysts, such as:
Comment: p. 5, line 179, figure 2: Clarify that the proportion in A to E refers to andiroba cake and soybean meal.
Response: Thanks for the comment, we added the missing information (Lines: 183-184).
Comment: p. 8, line 275: Change "conformation" to "configuration".
Response: Thanks for the suggestion. We performed the exchange of words (Line: 305).
Comment: p. 8, line 275: "cis" should be italicized.
Response: Thanks for the observation. We put the word in italics (Line: 305).
Comment: p. 8, figure 6: Clarify in figure 6 the meanings of the letters "a", "b" "ab" and "c'.
Response: Thanks for the comment, we added the missing information in the figure caption (Lines: 326-327).
Comment: p. 9, item 3.3, eq. (1): Clarify the meanings of "ε" and "ρ".
Response: We apologize for our mistake. We added the meaning of "ε" and "ρ". Also, we added the reference for the calculation (Lines: 385-387).
Comment: p. 9, line 341: What is the meaning of the acronym BOD?
Response: We have already included the meaning in the text. The meaning of BOD is biochemical oxygen demand (Line: 393).

Reviewer 3 Report
Even though the results of this paper is meaningful for lipase preparation and oil hydrolysis, the whole novelty was limited as a scientific paper. Maybe the authors need to reorganize the paper to emphasize the importance of this work.
1. The authors mentioned the high cost of oil hydrolysis, could you explain it clearly? Meanwhile, could you calculate the cost of the whole process of the lipase preparation in your work?
2. I personally think that the phrase “solid enzymatic preparation (SEP)” was not exactly accurate, especially in the expressions like “0.5 g of SEP”.
3. The authors emphasize the usage of andiroba oil cake and soybean meal as byproducts, my question is that if the contents as in Table 1 changed a little which should be normal during the industrial process, will the solid state fermentation and solid enzymatic preparation process change, especially the optimized operating conditions?
4. Some figures listed error bars, and some didn’t, can I know why?
Author Response
Dear Reviewer,
Thank you for your useful comments and suggestions on our manuscript. We have modified the manuscript accordingly and the detailed corrections are listed below. The manuscript was amended following the recommendations of each reviewer. The changes are marked with different text colors to facilitate visualization.
Reviewer comments:  
Reviewer 3
Even though the results of this paper is meaningful for lipase preparation and oil hydrolysis, the whole novelty was limited as a scientific paper. Maybe the authors need to reorganize the paper to emphasize the importance of this work.
Comment: The authors mentioned the high cost of oil hydrolysis, could you explain it clearly? Meanwhile, could you calculate the cost of the whole process of lipase preparation in your work?
Response: We appreciate the comment and recognize the difficulty of defining the economic viability of obtaining enzymatic extract and solid biocatalyst from low-cost agro-industrial by-products. A brief explanation of these obstacles was included in Lines 347-358.
Comment: I personally think that the phrase “solid enzymatic preparation (SEP)” was not exactly accurate, especially in the expressions like “0.5 g of SEP”.
Response: We replaced the term "SEP" to solid biocatalyst (Lines: 31, 37, 38, 40, 145, 258, 314, 315, 317, 319, 323, 326, 349, 405, 409 and 418).
Comment: The authors emphasize the usage of andiroba oil cake and soybean meal as byproducts, my question is that if the contents as in Table 1 changed a little which should be normal during the industrial process, will the solid-state fermentation and solid enzymatic preparation process change, especially the optimized operating conditions?
Response: We appreciate the opportunity to improve the discussion about the use of by-products and recognize the difficulty of defining the economic viability of obtaining enzymatic extract and solid biocatalysts from low-cost agro-industrial by-products. A brief explanation of these obstacles was included in Lines 347-358.
Comment: Some figures listed error bars, and some didn’t, can I know why?
Response: We apologize for our mistake. We have modified Figure 3 (Lines: 223-224) and added the error bars to the revised version. Regarding the other figures, the error bars are smaller than the symbol and so are not observed.

Round 2
Reviewer 1 Report
The necessary changes have been made, I have no additional question.
Reviewer 3 Report
I think this version is more clear for the expression of the significance of the work.